# Wastewater-Based Surveillance Reveals the Effectiveness of the First COVID-19 Vaccination Campaigns in Assisted Living Facilities

**DOI:** 10.3390/ijerph21091259

**Published:** 2024-09-23

**Authors:** Katherine I. Brenner, Bryan Walser, Joseph Cooper, Sunny Jiang

**Affiliations:** 1Samueli School of Engineering, University of California, Irvine, CA 92617, USA; kbrenne1@uci.edu; 2Pangolin LLC, 260 Southhampton Ave., Berkeley, CA 94707, USA; bryan.walser@gmail.com (B.W.); prospect76@me.com (J.C.)

**Keywords:** wastewater epidemiology, SARS-CoV-2, vaccine effectiveness, non-pharmaceutical interventions, assisted living facility

## Abstract

The COVID-19 pandemic has disproportionately affected vulnerable populations, including residents of assisted living facilities (ALFs). This study investigates the impact of non-pharmaceutical interventions (NPIs) and mass vaccination campaigns on SARS-CoV-2 transmission dynamics within four ALFs in Maricopa County, Arizona, United States from January to April 2021. Initial observations reveal a significant SARS-CoV-2 prevalence in Maricopa County, with 7452 new COVID-19 cases reported on 4 January 2021. Wastewater surveillance indicates elevated viral loads within ALFs with peak concentrations reaching 1.35 × 10^7^ genome copies/L at Facility 1 and 4.68 × 10^5^ copies/L at Facility 2. The implementation of NPIs, including isolation protocols, resulted in a rapid decline in viral loads in wastewater. Following mass vaccination campaigns, viral loads reduced across all facilities, except Facility 4. Facility 1 demonstrated a mean viral load decrease from 1.65 × 10^6^ copies/L to 1.04 × 10^3^ copies/L post-vaccination, with a statistically significant U-statistic of 28.0 (*p*-value = 0.0027). Similar trends are observed in Facilities 2 and 3, albeit with varying degrees of statistical significance. In conclusion, this study provides evidence supporting the role of NPIs and vaccination campaigns in controlling SARS-CoV-2 transmission within ALFs.

## 1. Introduction

There are more than 30,000 assisted living facilities (ALF) in the United States [1]. These facilities are largely affected by the coronavirus pandemic due to high concentrations of vulnerable residents in the same facilities. On 20 January 2020, the first case of COVID-19 was reported in the U.S., with the highest number of clinical cases and hospitalizations in the U.S. occurring in the period December 2020–January 2021 [2]. The CDC reports that by 15 October 2020, in the 39 states examined, 22% of ALFs reported at least one case of COVID-19 among staff and residents. Additionally, 21% of residents with COVID-19 died, compared with 3% in the general population with the illness [3]. Prior to the development of highly effective vaccines, global efforts to reduce the transmission of SARS-CoV-2, the causative agent of COVID-19, primarily involved non-pharmaceutical interventions (NPIs) including social distancing and isolation [4,5]. The arrival of messenger RNA (mRNA) vaccines starting in December 2020 has led to decreased COVID-19 cases and deaths in ALFs housing high-risk individuals [6]. Nevertheless, quantitative information related to the effectiveness of NPIs and vaccination campaigns in disrupting the transmission of SARS-CoV-2 has not been fully examined in ALFs.

Wastewater-based epidemiology (WBE) has been used to track the prevalence of various viral diseases such as poliovirus [7], Hepatitis A [8], and most recently SARS-CoV-2 [9,10,11,12]. Clinical and asymptomatic SARS-CoV-2 infections are frequently accompanied by viral RNA shedding in feces and urine, detectable in wastewater. Several clinical studies have demonstrated that shedding lasts at least 3–4 weeks after individuals are symptomatic [13], and thus, wastewater surveillance can be used to show cumulative viral shedding. SARS-CoV-2 RNA wastewater surveillance has been used on college campuses, cruise ships, and at the community level [9], and has been discussed as an approach to complement clinical testing by using accessible and comprehensive pooled samples in long-term care facilities [14]. Wastewater detection may serve as a possible early-warning surveillance of COVID-19 infections in areas of both high and low prevalence [15,16] and is a cost-effective alternative which can provide more rapid results than clinical testing under certain circumstances [17] With delays in clinical testing and a widespread inability to effectively find asymptomatic cases, wastewater provides accessible and aggregate samples to detect SARS-CoV-2 within a population [10]. Wastewater-based surveillance (WBS) at neighborhood and facility levels can provide anonymous and rapid feedback about the effectiveness and utility of infection control measures in the distal sewershed at that location, far from the central treatment plant, in order to preserve specific geographic information [18,19].

We report a case study of four ALFs in the greater Phoenix area, Arizona, monitored using “below-ground” sampling of wastewater combined with “above-ground” nasopharyngeal swab qPCR testing to rapidly localize, respond to, and monitor repeated threatened outbreaks of COVID-19 at the level of specific buildings, as well as specific care units housing a vulnerable subpopulation. The wastewater contained a pooled selection of contributions from each member of the population using the restroom facilities within the sampled locations during the designated collection period. The outcomes of this study demonstrated the usefulness of WBS in the understanding of the effectiveness of vaccination campaigns and non-pharmaceutical interventions (NPI).

## 2. Materials and Methods

### 2.1. Description of Study Sites

The four ALFs investigated in this study were chosen based on data availability and the support of the facility management for wastewater surveillance. All four facilities are classified as assisted living facilities, including one memory care facility, located in Maricopa County, Arizona (Figure 1). These facilities are among over 500 ALFs in Maricopa County, including at least 24 memory care units. Facility 1 and 2 are located on the same campus but are composed of two sub-facilities: a regular assisted living building (referred to herein as “Facility 1”) and a memory care building (referred to herein as “Facility 2”). Facility 3 and 4 both provide assisted living for seniors and are located on separate sites within 10 miles of each other.

Residents of the facilities are full-time residents, often with underlying health problems, needing assisted care. The age range and demographics of the sample population is not collected for privacy protection. Facility 1 offers studio, 1-bedroom, and 2-bedroom apartments. Facility 2 offers studio and 1-bedroom accommodations. Residents dine together in a communal dining room and socialize in common areas. Facility 2 is a memory care assisted living facility which provides housing and care for those with Alzheimer’s disease or other forms of dementia in a more contained environment. To protect the identity of these facilities, locations and other details of the facilities are intentionally omitted.

At the beginning of the monitoring program in January 2021, no residents or staff had been vaccinated for COVID-19. Facilities 1 and 2 have a combined 200-resident capacity, employing approximately 80 staff (60 full-time and 20 part-time equivalents). The occupancy at the time of monitoring was about 55%. Facility 3 and 4 each has a capacity of 120 and 130 residents, respectively, with over 30 staff members working at each facility. In all the facilities, only residents were housed in the facility. No staff members lived on-site of any facility during the study period between January and end of April 2021. Family visits to the residents on-site were prohibited during this period. The populations in the ALFs did not vary during the course of this study. The number of staff on campus varied slightly each day due to work schedules. However, at any given time, there were at least 30 staff members in each facility, and these staff members were reoccurring staff. Therefore, these isolated facilities provided a more controlled setting for wastewater surveillance in comparison with open communities with population fluctuations.

### 2.2. Wastewater Sampling

The wastewater monitoring was initiated on 5 January 2021, after consulting with facility management and environmental monitoring firms. Four manholes were selected to collect wastewater flow from each of the four ALFs. Separate sewer lines from each subsection allows for separate wastewater surveillance in the campus. ISCO 3750 composite samplers (Western Environmental Equipment Company Inc., Scottsdale, AZ, USA) were installed in each manhole using a below-manhole sling. A dye study was conducted on the first sampling day to confirm manholes were positioned correctly to collect wastewater flow from the particular populations involved. In the case of Facility 1 and 2, sewage runs from both facilities to the main sewer lines via manhole connectors but flows separately for a distance of several yards. This enables separate sewage sampling locations for separate resident populations.

The automatic samplers were programmed to collect an aliquot of 20 mL wastewater every 15 min per 24 h to form a composite volume. This setup was used to collect the maximum number of samples the automatic samplers could support within the 2 L total volume. Collection began on 5 January, and continued every Monday, Wednesday, and Friday throughout the months of January until the end of April 2021. A total of 168 wastewater samples were collected from four manholes over the study period, i.e., 42 samples from each facility. The flowchart in Figure 2 outlines the wastewater surveillance action plan developed in late 2020 before the initiation of the field program.

### 2.3. Sample Analysis for SARS-CoV-2

Wastewater sample analysis for SARS-CoV-2 was conducted off-site by GT Molecular (Fort Collins, CO, USA) using droplet digital PCR (ddPCR) protocols based on TaqMan reagents and EvaGreen fluorescent reporter, according to procedures reported by Miotke L. et al. [20]. Bovine coronavirus was used as an internal process control in each analysis. The wastewater sample was spiked with a bovine coronavirus at a known concentration to account for any virus loss during processing. QA/QC of molecular analysis was carried out as part of GT Molecular’s routine operation protocols. The results that were received from GT Molecular included viral copies per liter of wastewater and QA and QC reports. The lower limit of detection of the virus ddPCR assay was 10,000 copies/L, which was calculated based on <1 copy per ddPCR reaction corrected by the concentration factor used for most of the wastewater samples. A higher concentration factor was used for a subset of samples, which resulted in positive detection results in some samples being below the 10,000 copies/L detection limit.

### 2.4. On-Site Health Management Protocols

Prior to initiating wastewater monitoring, facility compliance with existing public health requirements was evaluated. This study was determined to be within the bounds of a strictly observational study to understand the impact of current standard practices and provide the additional information content available from wastewater pathogen detection. Subject-level informed consent was not required. The health management protocol also requires that investigators notify the relevant authorities of the facility management immediately upon detection of significant levels of SARS-CoV-2 in wastewater from specific facilities. Investigators are required to alert authorities as to the presence of a potential COVID-19 transmission among a vulnerable population and to provide relevant information to help with the prioritization of the scarce vaccine resources available at that time.

All ALFs actively participated in the clinical testing program during the study period. ALF residents and staff were tested at least once per week using reverse transcription quantitative PCR (RT-qPCR) of nasopharyngeal swabs by certified clinical labs. Staff that tested positive were remanded to their home for the minimum period of quarantine then required by CDC, being allowed to return only after a negative test result. Residents that tested positive were remanded to isolation in their unit, and allowed to leave only after a negative test was obtained after at least 10 days of isolation. Non-symptomatic residents were also tested at regular intervals. For both baseline and return testing, staff and/or residents were re-contacted until compliance was achieved, resulting in full testing within the subpopulation. Clinical test results were communicated, verified, and transmitted to the management of the ALF within 24 h after laboratory reporting. Since clinical testing results were only verifiable for Facility 1 and 2, only data from these two facilities were included in the analysis together with wastewater data. The results were reviewed for the impact of management actions on viral presence and inferred transmission (Figure 2).

To investigate the relationship between SARS-CoV-2 concentrations in wastewater and COVID-19 cases, a two-tailed Spearman’s rho rank correlation (r_s_) was computed. A *p*-value of <0.05 was considered statistically significant, indicating a meaningful correlation between the two variables.

### 2.5. Vaccination Campaigns

Vaccination campaigns at Facility 1 and 2 were initiated using the Moderna mRNA vaccine on 10 February 2021, and the second dose administered on 10 March 2021. A backup appointment date was provided for residents on 7 April 2021. The vaccination campaign at Facility 3 was conducted using the Moderna mRNA vaccine on 25 January 2021, and the second dose on 22 February 2021, with a third alternative vaccine date of 22 March 2021. The vaccination campaign at Facility 4 was conducted using the Moderna mRNA vaccine on 18 January 2021, and the second dose on 15 February 2021, with a third alternative vaccine date of 15 March 2021. The management of the facilities estimated that approximately 90% of residents and 60% of staff were vaccinated with their second dose by the end of March 2021. Residents and staff were considered fully vaccinated 2 weeks after their second vaccine dose. The timeline of vaccination campaigns at each facility are illustrated using vertical green dash lines in Figure 3. Seven-day new case rates were obtained from Maricopa County, Arizona Public Health Department.

### 2.6. Statistical Analysis

A Mann–Whitney U Test was used to assess the null hypothesis that vaccination does not lead to a significant reduction in viral levels within wastewater outflows. The wastewater data were binned into two independent groups: data collected before the completion of the second dose of vaccine (Jan to Feb 2021) were binned into “prior-vaccination”, while the data collected between March and April were binned into “post-vaccination”. A value of ½ LOD (5000 copies/L) is used for samples reported as non-detect results.

Spearman’s rho rank correlation was used to test the null hypothesis that wastewater SARS-CoV-2 concentrations are not correlated with the on-campus clinical cases. A *p*-value of <0.05 was considered as grounds for rejecting the null hypothesis. The Spearman’s rank correlation analysis was carried out using Microsoft Excel ™ version 2408. Due to the relatively short duration of this study and the large number of non-detect results, further statistical analysis using time series tools was limited. The observations were treated as functionally independent groups in the statistical analysis.

## 3. Results

### 3.1. SARS-CoV-2 RNA in Wastewater

The four facilities showed an intermittent but consistent presence of SARS-CoV-2 RNA in the wastewater. The highest quantitative SARS-CoV-2 RNA viral load from this study occurred at Facility 1 on 7 January (1.35 × 10^7^ copies/L), correlating with positive clinical tests among residents and staff. As per CDC guidelines, positive cases were immediately isolated. In the weeks following this significant viral load, on 21 January and 26 January, no detection (ND) of wastewater SARS-CoV-2 RNA was observed (Figure 3). Facility 2 showed a viral concentration of 4.68 × 10^5^ copies/liter in wastewater collected on 7 January, but viral loads dropped below detection limits in the following week (Figure 3).

To compare the wastewater signal with clinical cases on campuses, we aggregate the Facility 1 and 2 clinical case data since they are located on the same campus and share staff between facilities (Figure 4). The results show that the elevated level of wastewater signals in early January is largely due to infected staff. The removal of the infected staff from the campus results in an immediate drop in wastewater SARS-CoV-2 concentrations. Moreover, the isolation of infected residents also prevented the rapid spread of the infection, as shown by the sporadic resident cases. Comparing the aggregated clinical cases with aggregated wastewater data from Facilities 1 and 2 reveals that the wastewater viral loads are consistent with clinical cases in the periods prior to the vaccination program. A mismatch was observed on 15 February when five cases of staff infection were recorded but no wastewater signal was detected. The clinical record did not indicate if the infected staff were working on campus at the time of clinical testing. The wastewater data suggest the infected staff are absent from the facility.

The Spearman’s rho rank correlation analysis of wastewater and clinical data yielded a coefficient of 0.74, with a *p*-value of <0.01, suggesting a strong positive linear correlation between the wastewater SARS-CoV-2 and COVID-19 clinical cases that was statistically significant at the 5% level (Figure 4). We noted that time-series-based statistics would be better suited for time-series data analysis. However, the large number of non-detect results in wastewater and low number of clinical cases limited the application of segmented regression or autoregressive integrated moving average (ARIMA) in the data analysis. The Spearman’s rho rank correlation analysis provides a general agreement between wastewater data and clinical cases.

Similarly, Facility 3 observed a significant wastewater viral load on 12 January (6.07 × 10^4^ copies/L) corresponding to a clinical positive from nasopharyngeal testing of residents. The immediate isolation of the infected resident prevented the further spread of the infection on campus. Wastewater viral loads dropped in the later wastewater testing, validating the effectiveness of NPI (Figure 3). The wastewater viral loads in Facility 4 remained low during this study, except at the end of this study. There were no verified clinical cases reported on this campus. The high concentration of wastewater viral RNA at the final sampling date on this facility requires further investigation (Figure 3).

It is interesting to note that wastewater viral loads on ALF campuses trended with 7-day new case rates of Maricopa County, Arizona (Figure 3). Notably, the Facility 1 and 2 peaks in wastewater SARS-CoV-2 RNA correspond with the peaks in the averaged 7-day new cases. From an observational point of view, this study can offer insight into the interconnectedness of a population mostly isolated from the surrounding area, in terms of disease prevalence. The interpretation of these results should consider the temporal variations, with intermittent elevations, and the potential correlation with clinical cases within the individual facility. A very large number of samples were reported as non-detects, indicated by “−” below the red line of detection limit in Figure 3. In a few exceptions, viral concentrations (indicated by “+”) below the general detection limit were also reported when a higher concentration volume was used. Improving the detection limit by concentrating a larger volume of wastewater could further improve the rigorous of the data and statistical power.

### 3.2. Effectiveness of Vaccination Campaigns

To assess the effectiveness of the first COVID-19 vaccination program for the ALF through wastewater surveillance, the results of wastewater viral load data before and after full vaccination were compared (Table 1). In Facility 1, the mean viral load before vaccination was 1.65 × 10^6^ copies/L, which significantly decreased to 1.04 × 10^3^ copies/L after full vaccination, with a U-statistic of 28.0 and a *p*-value of 0.0027, indicating a significant reduction. For Facility 2, the mean viral load before vaccination was 3.12 × 10^4^ copies/L, decreasing to 149 copies/liter after full vaccination. This change was not statistically significant, with a U-statistic of 66.0 and a *p*-value of 0.17. Facility 3 showed a mean viral load of 7.38 × 10^3^ copies/L before vaccination, reducing to 918 copies/liter after full vaccination, showing a trend toward a reduction, but the change was not statistically significant (U-statistic = 72.0, *p*-value = 0.53). In Facility 4, SARS-CoV-2 was infrequently detected throughout the study period, with a mean viral load before vaccination of 3.00 × 10^3^ copies/L, increasing to 5.18 × 10^4^ copies/L after full vaccination. This high concentration detected in the post vaccination was due to a large spike from a single wastewater sample at the end of the study period. There was not a record to indicate the presence of clinical cases, but we cannot rule out asymptomatic infection among the post-vaccination population. U-statistics yielded a U-value of 23.5 and a *p*-value of 0.73, suggesting no statistical difference before and after vaccination in Facility 4. Since Facility 4 was the first to implement the first vaccine dose and only had six data points before vaccination, these data are skewed, leading to limitations of this analysis.

We also noted the large variability in the wastewater SARS-CoV-2 concentrations and the high number of samples that were below the detection limit, as indicated by the median interquartile range (IQR) values of zero in facilities. The large number of below-detection results prevents the further statistical analysis of the data trend.

## 4. Discussion

The findings of this study underscore the critical role of non-pharmaceutical interventions (NPIs) and vaccination campaigns in mitigating the spread of SARS-CoV-2 within ALFs. The observed effectiveness of NPIs, such as isolation and quarantine methods, is consistent with previous research indicating the potential of containment strategies in controlling outbreaks [22]. Most notably, our study demonstrates a rapid decline in viral loads following the implementation of such protocols, such as removing infected staff from coming to campus and isolating infected residents to prevent further spread, corroborating the effectiveness of such interventions in curbing transmission dynamics [23]. It is also important to note that the infected residents in quarantine remained at the ALF campus and still contributed to the viral load in wastewater. The decline in viral signals in wastewater does not necessarily represent the complete absence of viral genomes because the detection limit for wastewater assay was around 10,000 genome copies/L. Therefore, the highly dynamic wastewater data, especially from Facility 1, suggest it is a sensitive indicator for NPI actions. Moreover, the impact of vaccination campaigns on SARS-CoV-2 RNA wastewater viral loads presents promising insights into the broader efficacy of vaccination in vulnerable populations. This is particularly important because individual testing in this type of facility is often subjected to privacy concerns, stakeholder cooperation, and liability issues. Use of wastewater presents pooled data without the need for individual consent. The information retrieved from wastewater could be used to educate the residents to cooperate with individual testing and vaccine programs.

While our findings align with studies reporting significant reductions in viral shedding post-vaccination [24], the observed transient elevations in viral loads post-vaccination warrant further investigation. These fluctuations may be attributed to various factors, including the emergence of viral variants or incomplete vaccine coverage [25]. The integration of wastewater surveillance with clinical testing not only serves as a valuable early-warning system for outbreak detection but also offers insights into the effectiveness of vaccination strategies. Our study highlights the complementary nature of WBE in monitoring infection dynamics, as evidenced by its ability to detect asymptomatic shedding and track transmission trends within confined settings [26].

Overall, pathogen surveillance of wastewater can provide regular, automatic, and comprehensive assessment of the viral load circulating within a defined population. It can supplement or perhaps even guide and direct associated clinical testing programs, in order to rapidly identify, implement, and assess infection control strategies and specific management actions and health interventions, including vaccination when and as available, in the context of a particular delimited outbreak within a widespread on-going viral pandemic.

Specifically, with respect to the cost-effectiveness of using wastewater as a preliminary assessment to guide subsequent clinical testing, a simple calculation can demonstrate the cost saving for a facility. For example, if one considers all staff (77, including part-time employees) and residents (111) being tested via the facility’s wastewater outflow, and then using a relatively less sensitive clinical test (such as an antigen detection assay) only after noting a viral outflow burden greater than a given relevant level (in this instance, assuming that level to be 100,000 genome copies per liter), the total testing program for the four months under observation would have been less than ~USD 50,000, making this a relatively economical tool that can be utilized to alongside clinical testing to significantly improve the efficiency of laboratory resource utilization. Therefore, WBE should be incorporated into the ALF health management decisions for cost, equality, and privacy considerations. Researchers are mindful of the ethical implications involved in using this method to study population health. While individual identities remain anonymous through wastewater sampling, the potential for stigmatization of specific neighborhoods or communities persists. For ALFs, this issue is especially critical, as policy decisions must address privacy concerns and ensure that results are communicated responsibly to avoid unintended harm or bias.

Comparing our findings with the existing literature, our study contributes to a growing body of evidence supporting the utility of WBE in assessing the efficacy of public health interventions. Studies by Ahmed et al. (2020) and Rosa et al. (2020) have similarly demonstrated the value of wastewater surveillance in monitoring community-level transmission dynamics and guiding intervention strategies [10,27]. However, while these studies emphasize the utility of WBE in urban environments, our research extends this framework to vulnerable populations in ALFs, filling a crucial gap in the current literature.

Despite the strengths of our study, several limitations are acknowledged. The absence of observed clinical cases in some facilities limits our ability to statistically correlate the effectiveness of pharmaceutical interventions. Additionally, the lack of established threshold levels for wastewater-based detection poses challenges in interpreting viral load data accurately. Future research endeavors should aim to address these limitations by refining threshold limits and conducting longitudinal studies to assess long-term vaccine efficacy. Moving forward, continued research efforts are essential to refining surveillance methods to create targeted public health interventions.

The time course of changing viral loads in this assisted living facility was reflective of the local viral transmission conditions in the population being monitored, as reflected by similar results in the other monitored ALFs (as well as office blocks), and strategies such as wastewater pathogen detection in order to improve baseline clinical testing performance—and therefore improving signal-to-noise ratios—may be among the most clinically effective and economically efficient ways to improve pandemic management before the local availability of a vaccine. Once a vaccine does become more widely available, this technique may be used to help guide, direct, and encourage its adoption where needed most.

Furthermore, these data also indicate that vaccination not only creates a substantial impediment to the development of additional new cases but appears to interrupt viral shedding as well. Even so, questions remain about the nature and causes of the transient viral spikes observed after the essentially complete immunization of the facility, and the relative impact of changing populations of people being sampled in this manner versus the changing viral populations themselves, as variant populations were beginning to develop at this time. These additional questions may be addressed by sequencing viruses obtained specifically from the outflow of highly vaccinated populations, or by another type of analysis of these transient elevations.

## 5. Conclusions

This is one of the few studies reporting WBE on long-term care facilities and the only study examining the effectiveness of NPIs and the first COVID-19 vaccination campaigns on ALF campuses through WBS. The following conclusions can be made from the results of this study:Wastewater SARS-CoV-2 monitoring showed the effectiveness of NPIs such as isolation as well as vaccination within ALFs.WBE, when implemented correctly, can resulting significant cost savings compared to undirected clinical-only testing.The identification and separation of infected staff, as well as the isolation of infected residents, proved effective in containing SARS-CoV-2 in this situation, while a subsequent vaccination campaign resulted in the precipitous drop in viral loads to below the limit of detection, in the context of a complete cessation of observed clinical cases.Intermittent and transient elevations of SARS-CoV-2 occurred from time to time after the vaccination campaign, but with no reportable clinical cases, highlighting both the potential establishment of local immunity among the residents of this facility and the potential utility of further sequencing these residual viruses to determine if the variant status may be related to the persistence in outflows otherwise free of observed viral load.

## Figures and Tables

**Figure 1 ijerph-21-01259-f001:**
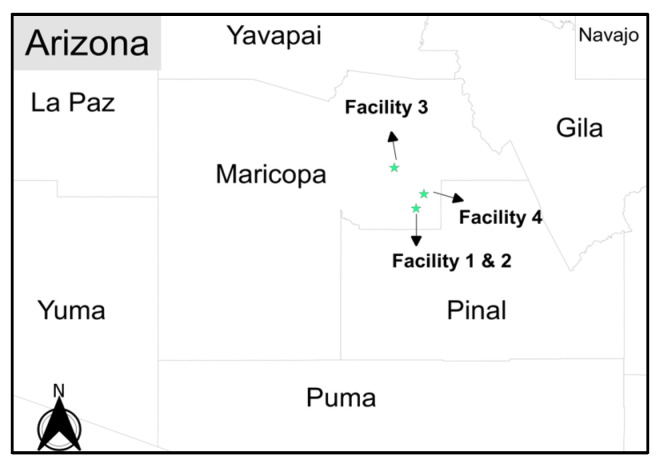
Locations of assisted living facilities investigated in this case study.

**Figure 2 ijerph-21-01259-f002:**
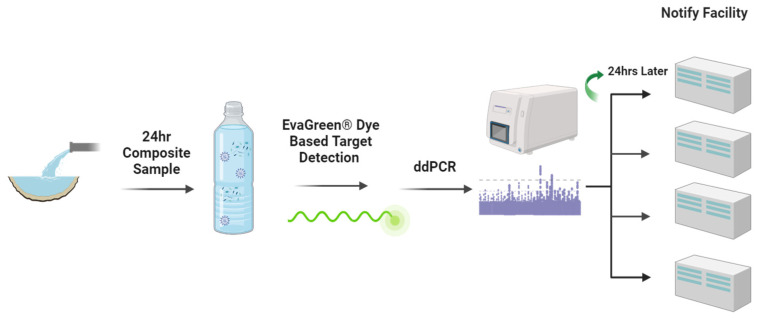
Flow chart of study design from wastewater sample collection and analysis to results being communicated to facilities.

**Figure 3 ijerph-21-01259-f003:**
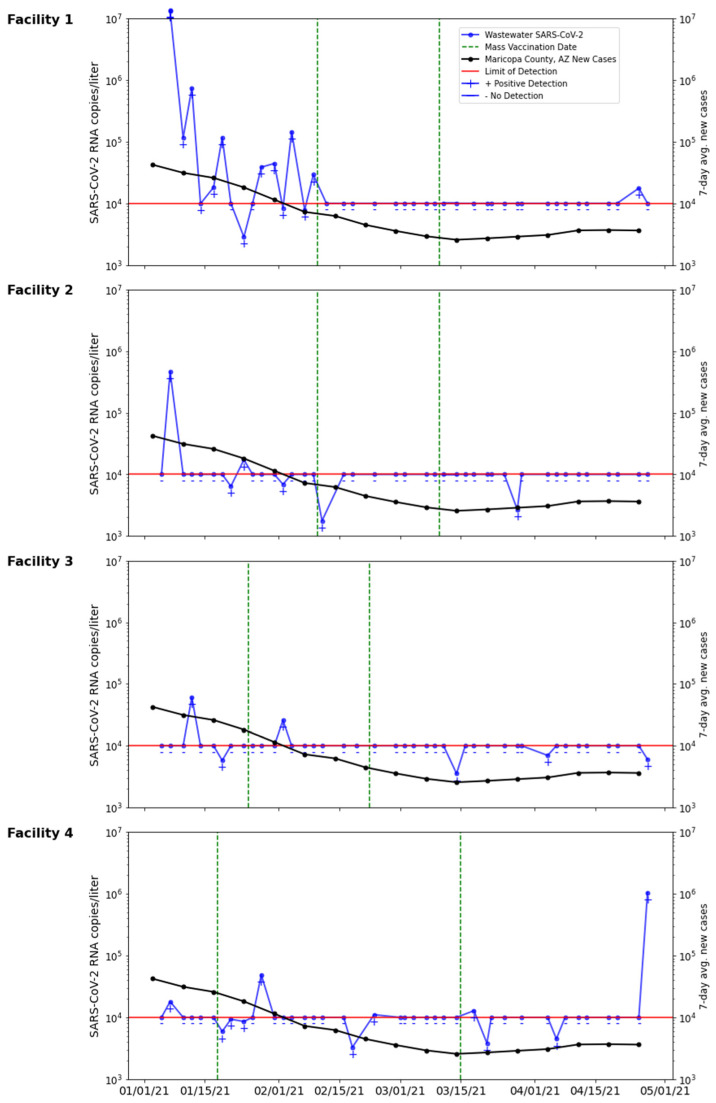
Wastewater SARS-CoV-2 RNA copies at each facility from 1/05/21 through 4/30/2021 over-laid with the 7-day average new case rate for Maricopa County, AZ, USA [21]. Vaccination campaigns at each facility are labeled by vertical green dash lines. The lower limit of SARS-CoV-2 RNA detection calculated using the general concentration factor is marked by a horizontal red line.

**Figure 4 ijerph-21-01259-f004:**
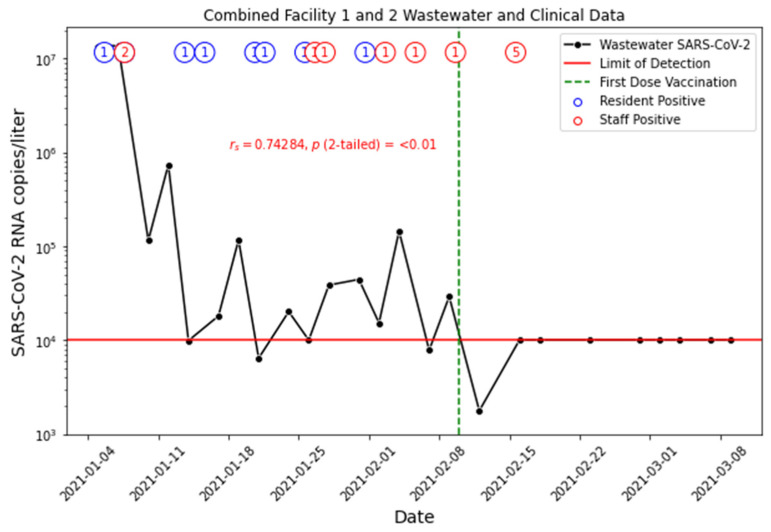
COVID-19 positive clinical cases among residents and staff in Facilities 1 and 2 overlaying the wastewater SARS-CoV-2 concentrations from two manholes. Clinical data are aggregated for Facilities 1 and 2 since staff are shared between facilities on the same campus. Number of residents with a clinical positive test is denoted as a blue circle with the number testing positive inside. Number of staff with a clinical positive test is denoted as a red circle with the number testing positive inside.

**Table 1 ijerph-21-01259-t001:** Comparison of wastewater SARS-CoV-2 concentrations before and after vaccination campaigns.

Facility	Before Vaccination (Copies/L)			After Full Vaccination (Copies/L)			U-Statistic	*p*-Value
	Mean	Median	IQR	Mean	Median	IQR		
Facility 1	1,757,130.2	33,860.0	115,878.0	1467.8	0.0	0.0	28.0	0.0027
Facility 2	31,176.6	0.0	1603.8	222.9	0.0	0.0	66.0	0.17
Facility 3	7384.3	0.0	0.0	918.0	0.0	0.0	72.0	0.53
Facility 4	3001.3	0.0	0.0	51,815.1	0.0	0.0	23.5	0.73

## Data Availability

The original contributions presented in the study are included in the article/Appendix A, further inquiries can be directed to the corresponding author.

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
