# Peer review of "Wastewater-Based Surveillance Reveals the Effectiveness of the First COVID-19 Vaccination Campaigns in Assisted Living Facilities"

_ijerph, 2024, doi:10.3390/ijerph21091259_

Round 1

Reviewer 1 Report

Comments and Suggestions for Authors

In this manuscript, the authors report results from a case study of conducting wastewater-based surveillance at four assisted living facilities in Arizona. The authors interpreted wastewater data with other information including county-level case counts and information about NPI and vaccination campaigns.  Overall, this is an interesting case study and adds to the literature on the utility of WBE to control SARS-CoV-2 transmission. However, due to the short study period and limited sample size, the evidence generated from this paper is not very strong. I also urge the authors to be more careful about the ethical considerations around WBE. Some of the statistical methods are not used appropriately and need revision.

Specific comments are as below:

Line 73. Figure 1. I am not sure how much value this map adds to the story. In addition, I fear a map of this level of spatial resolution could result in easy reidentification of the four facilities. A good rule of thumb is the small cell size rule of 11. How many ALFs are there in Maricopa County, AZ? If it is less than 11, then I am wondering if the county name should be masked to protect the privacy of the residents living there?

Line 82. Figure 2. Similar to the comments above, I don’t think aerial imagery of the four facilities adds to the scientific story. This map also increases the risk of reidentification.

Line 119, please include the exact number of wastewater samples. The authors made all the wastewater sample data available in the Supplemental Material, so they should have the exact number for the sample size.

Line 128, QA/QC, not QA/QA

Line 157. The authors should explain here which sites have clinical testing data available and which sites do not and why. I was expecting to see clinical testing data for all four sites, only until much later in the manuscript did I realize that this data is not available for all four.

Line 172. Independent Student’s t-tests are not appropriate because these samples are time-series and not independent from each other. The authors should consider alternative methods such as segmented regression or ARIMA.

Line 214. Figure 4. Consider using different symbols to represent detected wastewater samples and non-detects. Explain why there are different viral load for samples that were below the detection limit?

Line 217. Typo, should be red solid line, not red dash line.

Line 219. Figure 5. The correlation coefficient was listed but there is no detail about which two variables are the correlations calculated for and what statistical tests did the authors use to obtain the p-value?

Line 260. Table 1. Average is not the appropriate statistic to use to summarize highly skewed, non-normally distributed data. The authors should include the median and inter-quartile range to the table. And again, T-statistics and p-value listed here are not appropriate because these samples are from a time series and clearly not independent.

Line 273. The authors observed considerable variability in wastewater data, is the viral load standardized at all? Did the authors keep track of the overall flow rate during the 24 hours? Are there evidence that within the same facility, the same population contributed to the wastewater each day?

Line 280. The authors should not state “wastewater does not need stakeholder cooperation”. It is very important that as a community, WBE researchers recognize the ethical implications of this approach to studying population health, and take great care in interpreting its findings. Even though individual identity is anonymized through wastewater sampling, specific neighborhoods or communities could still be stigmatized if results are shared without consent.

Line 309-310. I find the argument of how WBE can save 90% of the spending a bit of a stretch. The authors just admitted that wastewater samples complement clinical tests and should be interpreted with other complimentary data. If WBE cannot replace clinical testing, where does the cost-saving come from?

Comments on the Quality of English Language

English is fine, a few typos were found, manuscript needs to be copyedited more carefully

Reviewer 2 Report

Comments and Suggestions for Authors

The authors present a well written, interesting report of transmission of COVID in assisted living facilities combining wastewater based epidemiology, clinical surveillance, and program implementation. However, there are some points that the authors should address prior to publication.

Major Comments

Was a population marker used in the wastewater analysis? This could account for the amount of human fecal matter in each wastewater sample. If not used, how did the population in the ALF change over the course of the study? The facility usage at the beginning of the study period is described, but not during the study period. Is there any reason to believe the facility usage would change during the week (for example, staff meetings on Fridays would require all staff to be present who would normally not all be in the building at the same time)? The likelihood of detecting a pathogen is dependent of the number of people in the catchment area, the authors need to address if the assumption of a constant number of people is valid.

Data is presented for all of Maricopa county during the study period. However, there is no description of how this data was obtained. This should be included in the Methods Section.

The statistical analysis section is incomplete. How were the average viral levels calculated (mean, median, mode)? What value was used when there were non-detects? Methods for the correlation in Figure 5 are missing.

Why is clinical data only presented from Facilities 1 and 2 even though the methods state that all ALFs participated in the clinical surveillance?

Why are there reported virus concentrations below the limit of detection? This appears in the text, Figure 4, Figure 5, and Table 1.

A measure of dispersion should be included in Table 1 (standard deviation, IQR etc.)

Bullet points two and three in the Conclusion are not directly supported by the original data presented in this manuscript. The original data to support these conclusions should be presented or these conclusions should be eliminated. (While the cost analysis is presented in the Discussion, if the authors wish this to be part of the original data, more detailed information would need to be presented in the Methods about the data collection/analysis etc for this statement.)

Minor Comments

Figure 1-a compass and scale would be helpful.

The description of the facilities in the text (lines 78-79) and Figure 2 are not consistent.

Line 85-Please clarify why the age distribution is unknown (ie: data not collected, not shared etc).

Line 95-Please be more specific as to the number of staff at Facilities 3 and 4, “over 30” is not sufficient.

Line 128-Should it say QA/QC?

Line 195-How were these compared? It is a visual comparison, if so, this belongs in the Discussion. If a statistical comparison was done, this needs to be described in the Methods and appropriate results (ie: p-values) presented.

Lines 207-212 would be more appropriate in the Discussion section.

Line 246-247 needs to be rewritten.

Round 2

Reviewer 1 Report

Comments and Suggestions for Authors

I appreciate the authors being open to my previous comments. The revised manuscript is much improved. All but one of my comments are addressed.

The only remaining comment is since the wastewater data are not normally distributed, the authors should be using the Mann-Whitney U Test (non-parametric version of the Student's t-test) and Spearman's rank correlation coefficient (non-parametric alternative of Pearson correlation coefficient) as the statistical test. 

Author Response

I appreciate the authors being open to my previous comments. The revised manuscript is much improved. All but one of my comments are addressed. The only remaining comment is since the wastewater data are not normally distributed, the authors should be using the Mann-Whitney U Test (non-parametric version of the Student's t-test) and Spearman's rank correlation coefficient (non-parametric alternative of Pearson correlation coefficient) as the statistical test.

Response: Thank you for your suggestions. We have replaced t-test analysis with a Mann-Whitney U Test. Spearman’s rank correlation coefficient is used as well. The changes did not influence the outcomes of the results. Please see tracked change in the revised manuscript.

Reviewer 2 Report

Comments and Suggestions for Authors

The authors have made appropriate revisions to their manuscript. No additional changes are suggested.

Author Response

Thank you!